# Healthy Food Pyramid as Well as Physical and Mental Activity in the Prevention of Alzheimer’s Disease

**DOI:** 10.3390/nu14081534

**Published:** 2022-04-07

**Authors:** Alina Kępka, Agnieszka Ochocińska, Małgorzata Borzym-Kluczyk, Sylwia Chojnowska, Ewa Skorupa, Małgorzata Przychodzeń, Napoleon Waszkiewicz

**Affiliations:** 1Department of Biochemistry, Radioimmunology and Experimental Medicine, The Children’s Memorial Health Institute of Warsaw, 04-730 Warsaw, Poland; e.skorupa@ipczd.pl; 2Department of Pharmaceutical Biochemistry, Medical University of Bialystok, 15-089 Bialystok, Poland; malgorzata.borzym-kluczyk@umb.edu.pl; 3Faculty of Health Sciences, Lomza State University of Applied Sciences, 18-400 Lomza, Poland; sylwiacho3@gmail.com; 4Department of Psychogeriatry, Independent Public Psychiatric Health Care Institution in Choroszcz, 16-070 Choroszcz, Poland; gosiap95@wp.pl; 5Department of Psychiatry, Medical University of Bialystok, 15-089 Bialystok, Poland; napwas@wp.pl

**Keywords:** Alzheimer’s disease, L-carnitine, diet, physical and mental activity

## Abstract

The ageing of the population is resulting in neurodegenerative diseases, including Alzheimer’s disease (AD), which are an increasing social, economic and medical problem. Diet and physical activity are now considered as important modifiable factors that help prevent or delay the development of AD and other dementia-related diseases. The pyramid of healthy nutrition and lifestyle is a way of presenting the principles, the implementation of which gives a chance for proper development and a long healthy life. The basis of the pyramid, in the first place, is physical activity. Our review of the literature in the PubMed database supports the hypothesis that complementary factors, such as proper diet, physical exercise and mental activity, have a positive impact on the prevention of neurodegenerative diseases. The nutritional recommendations for healthy adults primarily include the consumption of vegetables, fruits, cereals, legumes, vegetable oils and fishes. Therefore, the introduction of Mediterranean and Asian diets may reduce the risk of the neurodegenerative diseases associated with dementia, whereas dairy products and meat—the main sources of L-carnitine—should be consumed in moderate amounts. The aim of our work is to provide up-to-date knowledge about the appropriate dietary model and healthy lifestyle elements and their impact on good health and the long life of people.

## 1. Introduction

Along with lengthening of the human life expectancy, it is possible to observe increases in the number of the neurodegenerative diseases that may be related to age, reduced physical fitness or mental capacity, thereby leading to disability [1]. During a literature search, we found data indicating that some food components are important in the prevention of the Alzheimer’s disease (AD) [2]. It is well known that adequate food consumption and a properly balanced diet have a positive effect on the good health of patients with neurodegenerative diseases, especially in AD [2]. Contemporary medicine (especially psychiatry) is paying more and more attention to diet components and composition as an alternative to drugs [3]. Currently, an increasing number of physicians and nutritionists are focusing on the diet and the qualitative composition of the diet, which are an increasingly attractive alternative for the treatment of neurodegenerative diseases beyond pharmacological treatment [3].

AD, Parkinson’s disease and dementia are connected with the degeneration of the majority of neurons in the temporal and central lobes, as well as some parts of the frontal lobes and callosal gyrus [4]. A neurodegenerative disease, i.e., the degeneration and death of the neurons, in the first stage of the disease, begins in the allocora area and then spreads to the hippocampus, amygdala, thalamus, forebrain and numerous monoaminergic nuclei of the brainstem. The cerebral cortex shrinks, damaging the areas responsible for thinking, planning and memory. The white matter is subject to significant disappearance, flattening of bends and deepening of furrows, widening of the side chambers and chamber III as well as the enlargement of the subarachnoid space. The appearance of amyloid clusters and features of neurofibrillary degeneration are also observed, which are typical lesions of AD [4]. The reasons for AD pathological symptoms are the extracellular storage of insoluble amyloid-β proteins (fAβ), intracellular storage of tau protein (microtubule binding phosphoprotein) and activation of the immune system in the brain [5]. It is believed that other proteins such as α-synuclein and the TDP-43 protein may play a role in the pathogenesis of AD [6]. In AD, the impaired transmission of nerve signals has been demonstrated due to a decreased density of muscarinic and nicotinic receptors as well as the decreased concentration and accumulation of acetylcholine (Ach) in synaptic vesicles, which is the most important for the memory system [6]. Moreover, the genetic basis for AD has been proven. The gene responsible for the development of the Aβ plaques characteristic for AD has been identified on chromosome 21. Moreover, more than 200 mutations have been found within the amyloid protein precursor (APP), presenilin 1 (PSEN1) and presenilin 2 (PSEN2) genes. PSEN 1 and PSEN 2 are membrane proteins located mainly in the membranes of the Golgi apparatus and the endoplasmic reticulum, where they form the active site of the γ-secretase complex that contributes to the early onset of AD pathogenesis [7]. An important role is played by the polymorphism of the apolipoprotein E (ApoE) gene. People with the APOE E4 genotype may have symptoms of AD appearing in childhood [8]. Unlike ApoE4, the E2 isoform is believed to be protective and may delay the age of onset. β-amyloid accumulates in neurons leading to electron transfer chain (ETC) dysfunction, resulting in decreased ATP production. It also leads to an increase in oxygen and nitrogen radicals. The balance between oxidants and antioxidants is also disturbed and, therefore, oxidative stress occurs, and the cellular homeostasis of Ca^2+^ ions is disturbed. In addition, the hyperphosphorylation of the tau protein inside neurons disrupts the intracellular transport of other proteins, including neurotrophins, e.g., nerve growth factor (NGF), as well as brain-derived neurotrophic factor (BDNF). The described neurodegenerative changes lead to the death of neurons and loss of interneuronal connections. They result in a decrease in the level of transmitter substances, which consequently leads to clinical symptoms of AD. The first symptoms characteristic of AD include impairment of memory, attention, speech, visual-spatial orientation and apraxia of different forms (difficulty in performing simple, learned movements such as writing, reading, using cutlery, tying shoelaces). Other symptoms in AD include but are not limited to: agnophasia (visual and hearing impairment, psychomotor restlessness and the wandering symptom), agnosia (lack of sensory ability to recognise objects despite normal sensory function, sleep disorders and insomnia, delusions as well as hallucinations, seizures and myoclonus). Constantly worsening disorders created by AD make it impossible for these people to function without help in everyday life.

In addition to the genetic predisposition for AD disease, the impact of other lifestyle-related risk factors on the development of AD is also emphasised. It was reported that a low level of education, obesity, depression, sleep disorder, hypertension, cerebrovascular diseases (most commonly reported risk factor), diabetes, smoking, dyslipidemia (elevated circulating cholesterol levels are capable of compromising integrity of the blood–brain barrier) and a sedentary lifestyle can contribute to the development of AD disease [9].

Among the new methods of preventing and treating AD disease are nutraceuticals, which play a key role. They are nutrients whose deficiencies may initiate or accelerate the development of pathological changes within the brain structures. A review of the literature still does not unequivocally allow formulating a set of nutritional recommendations that may have an impact on the prevention of AD disease. However, it has been clearly stated that a diet rich in certain ingredients can have a positive impact on the reduction in the risk of developing many neurodegenerative diseases. The most important compounds include uridine nucleotide, choline and PUFAs, which increase the level of phospholipids in the brain [9,10]. In addition, folic acid, vitamins B12, B6, C, E and selenium have now been shown to have a protective effect on cognitive function and to potentially reduce depressive symptoms during ageing, and their deficiencies, including high levels of homocysteine, are associated with cognitive impairment in the elderly [9,10,11]. A clinical trial of LipiDiDiet with Fortasyn Connect (a specific combination of a number of nutrients, i.e., omega-3 fatty acids, choline, uridine monophosphate, phospholipids, antioxidants and B vitamins) has shown that in people in the early phase of Alzheimer’s disease (the phase preceding dementia), consumption of a nutritional drink once a day leads to a significant stabilization of cognitive and functional parameters and reduces the degree of brain atrophy [12]. The share of antioxidant enzymes (superoxide dismutase, catalase and glutathione peroxidase) and non-enzymatic antioxidative substances (vitamin A, C, E, carotenoids, polyphenols, glutathione) in the diet of the elderly is an important factor in the prevention of AD disease because they play a protective role in preventing oxidative stress that affects the performance of the nervous system. In addition, the participation of many other natural compounds that occur in spice plants (curcumin, rosmarinic acid, ursolic acid, piperine, capsaicin, green and black tea extracts, coffee, cocoa) prevents neurodegenerative diseases [13], as well as the resveratrol–polyphenol present, among others, in grapes (mainly in the skin), red wine and peanuts [14]. It has been shown that the natural therapeutics of anti-oxidative and anti-inflammatory compounds evidently play a crucial role in neuron protection. Of these, the most promising are caffeine, trigonelline, shogaol, curcumin, resveratrol, baicalein, wogonin, ginsenosides, tanshinones, withanolides, picrosides, parthenolide, cannabinoids, Devil’s claw and white willow bark, including Chinese formulations Renshen Shouwu and Shengmai San [13]. Therefore, the nutritional recommendations for older people include primarily eating vegetables, fruits, cereals, legumes, vegetable oils and fishes, whereas dairy products and meat should be consumed in moderate amounts. Red meat and other products rich in choline, lecithin and L-carnitine are potential sources of the harmful toxic trimethylamine N-oxide (TMAO) that can be converted into the carcinogenic N,N-Dimethyl-N-Nitrosoamine (NDMA). Recent studies point to the potential contribution of gut-microbiota-derived production of TMAO from the metabolism of dietary choline and L-carnitine, which has been associated with an increased risk of major adverse events in humans, including neurological disorders [15].

Many studies have proven that L-carnitine (LC), which should be supplied with the diet (red meat and dairy products) [16,17,18], or delivered as a dietary supplement, plays an important role in preventing AD [19,20]. L-carnitine acetylation increases its hydrophobicity that permits acetyl-L-carnitine (ALC) to cross the blood–brain barrier. The results of many studies suggest that ALC increases the level of the neurotransmitter acetylcholine (ACh) in the brain, being a donor of acetyl groups in its synthesis [19,21]. ALC treatment increases (~16%) the expression of the neurotrophin receptor p75-mRNA level. This receptor plays important roles in regulating the amyloid beta-peptide (Aβ) metabolism in the brain, and in association with dissimilar decreases in levels of soluble amyloid beta peptide products (sAβ40 and sAβ42), with no change in the level of soluble amyloid precursor protein α (sAPPα), suggest an effect on Aβ peptidase activity [22]. The experimental data of Zhou et al. [23] showed that ALC supplementation in rats suppressed the Aβ peptides phosphorylation of amyloid precursor proteins (APP), which may indicate the reduction in Aβ in the brain [23]. Thus, ALC has a positive effect on APP in preventing amyloid plaque accumulation and a positive effect on the preservation of synaptic functions [23]. In addition, ALC protects muscarinic receptors and restores the activity of the nervous growth factor (NGF), which is extremely important for the proper functioning of neurons. Moreover, ALC modulates, directly or indirectly, the N-methyl-D-aspartate (NMDA) receptor, which introduces Ca^2+^ into cells because the excessive influx of Ca^2+^ induced by glutamic acid, which stimulates NMDA receptors, disturbs the homeostatic balance of neurons [24]. Furthermore, it has been reported that carnitine supplementation essentially elevates the levels of noradrenaline, adrenaline, serotonin and dopamine in the cerebral cortex, hippocampus and striatum in older rats [25].

In addition to L-carnitine (especially ALC), the other important factors affecting the quality of life of older people are lifestyle and regular and appropriate physical, mental and social activity, as well as a balanced diet, which allow older people to maintain health and a good quality of life without the features of infirmity [26]. In contrast, physical inactivity and a sedentary lifestyle are the main causes, alongside chronic diseases, of decline in physical and mental fitness for older people.

## 2. Food Pyramid

The pyramid of healthy nutrition and lifestyle is a way of presenting the principles, the implementation of which gives a chance for proper development, intellectual and physical activity and long life in health. By following the nutritional recommendations for adults, it helps to ensure good health and reduce the risk of dementia, whereas improper eating habits can result in distant deteriorative health effects. The basis of the pyramid, in the first place, is physical activity. The higher the level of the pyramid, the smaller the amount and frequency of consumed products from a given food group. One of the principles of a healthy diet is regularly eating foods that are varied and from different food groups (Table 1).

To ensure good health, we should limit the consumption of animal and vegetable fats, e.g., butter, palm oil contained in cakes, candy bars and fatty meat. We should replace animal fats with vegetable oils, e.g., rapeseed oil or olive oil, but consumed in small amounts and preferably in raw form, as an addition to salads or other dishes. Vegetable fats provide unsaturated fatty acids, among which especially monounsaturated and polyunsaturated fatty acids from the omega-3 family are very beneficial for our body. For short-term frying, it is preferable to use rapeseed oil or olive oil (the monounsaturated fatty acids they contain are more resistant to harmful changes under the influence of high temperature). For deep frying, saturated and monounsaturated fats are recommended (lard, clarified butter, coconut oil—although when consumed in excessive amounts, they are just as harmful as eating animal fats). Instead of traditional frying or baking dishes, other meal preparation techniques can be applied, e.g., baking in aluminium foil, sleeve, grilling, cooking, steaming and braising without frying the food first. One of the nutritional recommendations is to increase the consumption of vegetables and fruits in the right proportions: three-quarters vegetables and one-quarter fruits because the macronutrients contained in them generally protect the body against cognitive impairment and dementia, including AD [27].

We should eat whole grains. In addition to whole-grain bread, e.g., wholemeal, graham and whole-grain brown rice is recommended, as well as wholemeal noodles and cereals, such as buckwheat and barley. The above-mentioned products contain more vitamins, especially from the B group (thiamine, riboflavin, niacin, folic acid), minerals (magnesium, zinc) and dietary fibre than processed food.

Do not forget to consume milk and milk products: at least 2 large glasses of milk every day, which can be replaced by yogurt, kefir and partly by cheese (Table 1). Rennet cheese should be consumed less often and in a smaller quantity. It has been proven that poorer cognitive function and an increased risk of vascular dementia are associated with lower consumption of milk or dairy products. In contrast, the elderly should limit the consumption of full-fat dairy products because full-fat dairy products can lead to a decrease in cognitive function [27].

Two or three times a week, we can eat meat and its products. Choose lean types of meat, preferably of poultry origin, e.g., turkey, chicken and lean red meat (beef and pork) in smaller quantities, and on other days of the week (minimum twice a week), eat roasted or cooked fish (salmon, tuna, herring, mackerel, cod) and dishes from legumes (peas, beans, lentils, soybeans) or eggs.

It is recommended to limit the consumption of sugar and sweets, which can be replaced with fruit and nuts (hazelnuts, pistachios, cashews, sunflower seeds, pumpkin, if not roasted or salted). It is advisable to avoid adding any more salt. The salt contained in food and salting together should be consumed in an amount of not more than 5 g per day, i.e., an approximately flat teaspoon. Instead of salt, you can use herbs: rosemary, oregano, thyme, basil, turmeric, garlic, ginger, cinnamon—they have valuable ingredients and improve the taste of food.

The issue regarding alcohol consumption, even in moderate amounts, is still debatable. There are no precisely formulated recommendations for safe alcohol consumption [28]. As a reminder, one standard serving of alcohol is 10 g 100% alcohol, which is contained in about 250 mL of beer (5%), 100 mL of wine (12%) or 30 mL of vodka (40%). Drinking even small amounts of alcohol at the level of low risk of harm, from the toxicological point of view, is risky, but in most cases, a one-time consumption of small doses (10–50 g/day for men, 5–25 g/day for women) does not cause significant damage to both health and psychosocial functioning of a person consuming alcohol [29]. However, it appears that in preventing the risk of dementia in AD disease, light to moderate alcohol consumption may be associated with a reduced risk of developing AD, whereas, in the case of incidence of vascular dementia only, current evidence indicates that alcohol only has a protective effect [33]. Current MIND diet recommendations [30] include the consumption of wine (especially red), but no more than one glass a day, or beer (non-alcoholic beer) [31] in moderate amounts [18].

The food pyramid includes fluid (water, tea, coffee) intake, at least 1.5–2.0 litres per day. Total water consumption includes water, water in beverages and water in food products. Let us start the day with a glass of water right after waking up. During the day, drink water often and slowly, but in small sips. That way, you can hydrate the body most effectively. Water should make up the majority of the fluids consumed. You can drink diluted fruit and vegetable juices (preferably freshly squeezed) or infusions, e.g., from mint, chamomile and sage. It is not recommended to drink larger amounts of boiled and sparkling water.

It is also very important to remember dietary supplements, especially for older people who are over 60 years old and of decreased ability to absorb vitamins and minerals taken with food. Dietary supplements for seniors should, therefore, strengthen their body and support metabolism, but they cannot replace a rational menu. Substances used to improve mitochondrial activity include L-carnitine, acetylcarnitine, coenzyme Q10, mitoquinone mesylate, N-acetylcysteine, lipoic acid, sodium pyruvate, omega-3 fatty acids (especially DHA) and vitamins C, E, K_1_ and B. In age-related diseases, a person suffering from Alzheimer’s disease should include all the most important nutrients in the right proportions: carbohydrates, proteins, fats, vitamins and minerals [10,11].

## 3. A Diet to Prevent Alzheimer’s Disease

The best effects of AD prevention occur when the dietary intervention takes place before the first prodromal symptoms appear, i.e., when the number of synapses, nutritional status, cognitive level and neuropathological changes in the central nervous system are compensated. Research shows that first prodromal AD symptoms occur at around the age of 50 years. However, it seems more realistic to take action against AD when the first symptoms appear, i.e., around the age of 60–65 years. In turn, the ideal age for assessing the results of AD prevention is the age of 70 years. Researchers carrying out preventive research propose to include, in AD prevention, a population of people between the ages of 50 and 70 without signs of cognitive impairment but with risk factors such as hypertension, hypercholesterolemia and obesity. Interventions would be multidirectional, both pharmacological and non-pharmacological, and the real time of their implementation should be at least 4–8 years [34].

In the prevention of neurodegenerative diseases, one of the main goals is the implementation of an appropriate diet, and properly balanced nutrition can prevent AD and support the pharmacological treatment of elderly people. Changes in daily habits that can protect against dementia, including AD, are primarily not smoking, maintaining proper blood pressure, optimal cholesterol, glucose, homocysteine levels, preventing overweight and obesity, reducing stress, mental gymnastics and physical training—these are important elements in the prevention of AD. Another element of prophylaxis against neurodegenerative diseases is the introduction of certain nutrients into the diet or eating according to a given nutritional model, which may be helpful in neurodegenerative disease prevention and may support the basic treatment of people with AD. It is believed that including omega-3 fatty acids, antioxidant vitamins, B vitamins, folic acid, plant polyphenols, fish and vegetables in the diet, and limiting the amount of fried meat, may reduce the risk of developing AD [35]. Meals prepared with heated, fried or irradiated food products involve reactions of the aldehydes and amino group (Maillard reaction), thereby substantially speeding up the occurrence of AD [36].

### Mediterranean Diet

Increased risk of AD and dementia is associated with dietary patterns high in saturated fat and simple carbohydrates, while diets high in mono- and poly-unsaturated fats, vegetables, fruits and lean proteins are associated with reduced risk. With overall health and well-being in mind, switching to a Mediterranean-style diet is not only a prudent lifestyle choice, but is also a scientifically proven issue that can benefit the prevention and treatment of many diseases. An association between the consumption of a Mediterranean diet and a reduction in the incidence of Alzheimer’s disease has been demonstrated, raising the prospect that the Mediterranean diet may be used as a modifiable risk factor for protection against AD [37]. Researchers indicate that the Mediterranean diet, i.e., eating fruits with a low glycaemic index, vegetables with low starch content, whole grains, nuts, pulses, fishes (especially sea fishes: halibut, herring, mackerel, sardines), plant oils (olive, colza, linen, sunflower) and limited consumption of red meat but regular low to moderate alcohol consumption (mainly wine with meals) may delay the development of AD [37]. In contrast, it limits and recommends the consumption of low or average consumption of dairy products and poultry [38]. The Mediterranean diet is a way of eating and lifestyle at the same time and is considered as one of the healthiest. It is well known that following the Mediterranean diet can prevent many diseases: obesity and diabetes, cardiovascular diseases, cancer and neurodegenerative diseases. Research by Abbatecola et al. [39] showed that in people on the Mediterranean diet, the risk of mild cognitive impairment (MCI) was reduced by 28% and the risk of developing AD decreased by 48% compared to people who did not use this diet. A new model of nutrition has been developed, which is the MIND diet (Mediterranean-DASH Intervention for Neurodegenerative Delay) composed of the DASH (Dietary Approaches to Stop Hypertension) diet and the Mediterranean diet. The MIND diet aims to improve the functioning of the nervous system and the work of the brain. It is used in the prevention of AD and neurodegenerative brain diseases. A growing body of evidence supports a multicomponent intervention to alleviate cognitive impairment. Adherence to the MIND diet and increased physical activity have been shown to be associated with a reduced risk of dementia [30]. Fish, cheese and yogurt should be moderately consumed, while meat should be rarely consumed as a part of complex dishes. Wine and beer (non-alcoholic beer) during main meals are also some of the components of the Mediterranean diet. Beer consumption, and its content of bioavailable silicon, reduces the accumulation of aluminium in the body and brain tissue and lipid peroxidation and protects the brain against neurotoxic effects through the regulation of antioxidant enzymes [31]. The MIND diet emphasises the consumption of 15 groups of products, 10 of which are recommended to be eaten as often as possible. These are: vegetables, nuts, berries, legumes, whole grains, poultry, seafood and fish and mainly olive oil as well as wine/beer. The other five groups of products should be very limited: red meat, fatty cheeses, margarine and butter, sweets and cakes as well as fried and fast-food products [18].

The Mediterranean and Asian diets are very similar to each other and are considered the healthiest. The traditional Asian diet is characterised by a significant predominance of plant products over animal products [40]. The products commonly eaten in Asia include, first of all, rice and other cereal grains, as well as potatoes (including sweet potatoes), pasta, fruits and vegetables, legumes, soybeans (tofu, soy noodles, soy sauce), vegetable oils (sesame, soy), seeds, spices and tea. The consumption of fat, mainly animal fat and dairy products, is low. The proportion of fish is generally low to moderate, with the exception of those living in coastal regions. A healthy and balanced meal of plant-based foods and drinks such as green tea, vegetable oils, red wine, fruit and herbal spices has a beneficial effect on the prevention of amyloid diseases such as Parkinson’s, prion and Alzheimer’s diseases [40]. Actually, some foods or food groups traditionally considered harmful such as eggs and red meat have been partially rehabilitated, but there is still a negative correlation of cognitive functions with saturated fatty acids. A protective effect of elevated fish consumption and a high intake of monounsaturated fatty acids and polyunsaturated fatty acids (PUFA), particularly n-3 PUFA, against cognitive decline has been confirmed [39]. A study by Martinez et al. evaluated the effects of a Mediterranean diet on cognitive function in men at high vascular risk. This multicentre, randomized study showed that following a Mediterranean diet supplemented with additional extra-virgin olive oil or mixed nuts significantly improved cognitive function compared with a low-fat diet in the control group [41].

## 4. Food Products with Beneficial/Adverse Effects on Health

### 4.1. Advanced Glycation End Products (AGEs)

AGEs are a group of complex and heterogeneous compounds that are divided into colourless, non-fluorescent pre-melanoidins and colourful, fluorescent melanoidins. The former group includes glyoxal, methylglyoxal, 3-deoxyglucosone and carboxymethyl-lysine, while the latter one includes pentosidine. They are naturally present in unprocessed raw animal products, and heat treatment initiates the formation of new AGEs in these products. Grilling, baking and frying, in particular, are responsible for the formation of advanced glycation end products. The pathological effects of AGEs are related to their ability to promote oxidative stress and inflammation by binding to receptors on the cell surface, altering their structure and function. The accumulation of advanced glycation end products has been shown to play a role in the development and progression of age-related diseases [42]. AGEs, and especially AGE-2 derived from glyceraldehyde, show significant toxicity in cortical neuronal cells, the presence of which has been demonstrated in AD patients. AGE-2 may be involved in the development of dementia caused by the loss of cerebral pericytes in vascular dementia and neuronal cell apoptosis in AD [43].

A high concentration of AGEs is contained in products containing sugar, processed meat and processed dairy products; food containing trans fats (margarines, creams, mayonnaise); and highly fried products (fried potatoes, crisped cakes, pizza, etc.). The least concentration of AGEs is contained in natural products as well as in raw and unprocessed food, i.e., fresh fruits (strawberries, raspberries, blueberries, blackberries, cherries, currants and avocado); vegetables (cabbage, tomatoes, carrots, spinach, broccoli and kale); seafood (fatty fish, shrimps, clams, lobsters and squid), and briefly heat-treated products: slow-boiled, steamed and processed at lower temperatures [18]. Appropriate dietary supplementation, i.e., lipoic acid, glucagon-like peptide-1 (GLP-1), and physical exercise may be an effective method to reduce protein glycation and oxidative damage to cells and tissues. It has been suggested that the blockade of RAGE (receptor for advanced glycation end-products) may also limit the consequences of protein glycation [44].

### 4.2. Fatty Acids

A correctly composed diet rich in polyunsaturated fatty acids from families of omega-3 and omega-6 and monounsaturated fatty acids (MUFA), as well as antioxidative vitamins, reduce the risk of AD development. Epidemiological studies indicate that increased consumption of foods containing polyunsaturated omega-3 (α-linolenic—ALA; stearidonic—SDA; docosapentaenoic—DPA; docosahexaenoic—DHA; eicosapentaenoic—EPA acids) and omega-6 (gamma linolenic—GLA; linoleic—LA; and arachidonic—AA acids) as well as MUFA (oleic-, palmitoleic-, erucic acids) and antioxidant vitamins (A and its isomer β-carotene, E, C) has the ability to prevent the adverse effects of free oxygen radicals and may also reduce the risk of developing AD [33]. Many epidemiological studies have been conducted on the myriad health benefits of omega-3 polyunsaturated fatty acids. Vegetable oils, nuts and seeds are a rich source of omega-3 fatty acids, especially alpha-linolenic acid. In turn, EPA and DHA are found in fatty marine fish such as herring, mackerel, halibut, salmon, menhaden and seafood, algae and the blubber of marine mammals such as seals and whales. Recently, oils derived from algae, mushrooms and unicells have become popular as sources of long-chain omega-3 fatty acids. As a result of the ageing of the organism, the activity of the ∆-6-desaturase enzyme decreases with age, leading to the inhibition of the synthesis of DHA and an increased risk of disturbances in the functioning of the central nervous system in the elderly. Therefore, an adequate intake of omega-3 fatty acids or supplementation, especially DHA, is important in the elderly [45]. To achieve a proper balance in the body, you should limit the intake of foods rich in omega-6, such as processed meat and dairy products, as well as fast food and ready meals. The ideal ratio between omega-6 and omega-3 should be from 1:1 to a maximum of 2.5:1. The contemporary diet is rich in omega-6 fatty acids, and the ratio to omega-3 is as high as 15:1. As a result, the excess omega-6 has a pro-inflammatory effect and is considered as one of the main causes of civilisation diseases. It is recommended to consume natural fish or algae oil as often as possible, and at least 1–2 fish meals per week to ensure an adequate supply of omega-3 fatty acids [18].

### 4.3. Milk and Dairy Products

Regarding the risk of cognitive decline/cognitive disorders in adults and the consumption of milk and dairy products, the relationship cannot be clearly established. The evidence is too weak to draw firm conclusions. Poorer cognitive function and an increased risk of vascular dementia have been found to be associated with the lower intake of milk or dairy products, which are valuable sources of complete protein, calcium, potassium, phosphorus, vitamin A, some B vitamins and probiotics in fermented products. In addition, the consumption of full-fat dairy products, and/or dairy fats, may also be associated with worsening cognitive function in older adults. On the other hand, other studies have found adverse effects of full-fat dairy products (whole milk, dairy desserts and ice cream) on cognitive function in older adults [46,47]. It has been suggested that the consumption of milk > 1 glass/day at midlife may be associated with a greater rate of cognitive decline over a 20-year period. A significant inverse relationship has been observed between higher milk and dairy intake and reduced the risk of AD, but studies are limited to the Japanese population only [48]. Studies for the relation between dairy products and cognitive decline are contradictory, and it probably depends on the type of dairy product and the quantity ingested. There are also positive effects of dairy product intake, e.g., in the prevention of sarcopenia, especially with a high consumption of low-fat milk and yogurt. In conclusion, the available scientific evidence is insufficient to clearly assess the positive or negative impact of milk or dairy intake on cognitive decline and Alzheimer’s disease risk. Therefore, in our opinion, low-fat milk and dairy products such as low-fat yogurt and skimmed cottage cheese should be consumed because the consumption of full-fat dairy products may impair cognitive function in the elderly.

### 4.4. Alcohol

The relationship between AD, dementia and alcohol use/abuse has been the subject of many studies with varying and conflicting results. It is still unclear whether light to moderate alcohol consumption can lead to dementia or whether alcohol consumption can reduce the risk of developing AD [49]. Numerous published data confirm that low to moderate alcohol consumption, including wine (part of the Mediterranean diet), which contains numerous polyphenols which have protective effects, including minimising the effects of oxidative stress, inhibiting β-amyloid deposition, significantly reduces the risk of dementia and the risk of AD [50]. However, excessive amounts of ethanol increase the accumulation of Aβ and tau protein phosphorylation, contributing to the development of Alzheimer’s disease. Observational studies indicate that high alcohol consumption leads to a deterioration of cognitive and executive functions and leads to alcoholic dementia. The link between alcohol consumption and cognitive decline is believed to be “J” or “U” shaped [51]. In contrast, light/moderate alcohol consumption is associated with a reduced risk of dementia in individuals aged 55 years or older [52]. Regarding the type of alcoholic beverage, the Ritinberg study [52] found no difference between wine, beer or liquor. It is believed that beer (which is a component of the Mediterranean diet) and its ingredients (carbohydrates, protein/amino acids, minerals, vitamins and other compounds, such as polyphenols) exert a beneficial effect in the prevention of Alzheimer’s disease. Regular consumption of beer, or low-alcohol or non-alcoholic beer, can prevent AD and other neurodegenerative diseases as it effectively reduces the accumulation of aluminium in the body and also alleviates the mineral imbalance in the body and the brain and the pro-oxidative and pro-inflammatory effects caused by aluminium [31]. While the Luchsinger study [53] found that only wine (particularly red) has the strongest protective effect because resveratrol, a sirtuin 1 activator and other polyphenols present in the grapes of red wine reduce Aβ plaque burden and improve cognitive function. It has been noted that resveratrol actually reduces the levels of Aβ40 and Aβ42 in cerebrospinal fluid, but at the same time, it accelerated brain atrophy [54]; further studies are needed to confirm these results. Surprisingly, heavy drinking in late life has no effect on dementia risk compared to non-drinkers [52,55], but heavy drinking in adolescence is associated with damage to the prefrontal cortex and the hippocampus, and with neurocognitive dysfunction, it increases the risk of alcoholic dementia, similar to AD [49]. Importantly, alcohol misuse is also associated with a number of other disorders. Many publications state that ethanol and its metabolites not only have neurotoxic effects but directly exert toxic effects on the mucous membranes of the mouth (including periodontium), oral cavity, throat, oesophagus, liver, stomach, pancreas, kidney and can cause lung cancer (especially in cigarette smokers) and that regular drinking (even in small amounts) can lead to alcohol dependence [56,57].

Many studies support the benefits of low to moderate alcohol consumption in preventing AD [52]. However, these results should be considered insufficient to suggest that long-term abstinence should consider alcohol consumption in their diet to prevent AD risk. In addition, other risk factors (in addition to binge drinking), such as smoking or abusing other substances, can play an important role in the development and progression of many diseases, including neurodegenerative diseases. We believe that light to moderate alcohol consumption may be important in preventing AD.

## 5. Functions of L-Carnitine in the Brain

L-carnitine (2-hydroxy-4-trimethylammonium butyrate) (LC) plays many important roles in the intracellular metabolism of the body, with the most important one being a contributor to cellular energy metabolism [58]. LC is actively transported to the brain through the blood–brain barrier by the two transporters: OCTN2(*SLC22A5* gene), a Na^+^-dependent transporter that is present in brain endothelial cells, and ATB^0,+^, a Na^+^-, Cl^−^ -dependent amino acid (neutral and basic) transporter expressed in the hippocampus, with low affinity for carnitine. OCTN2 is found in the cells forming the blood–brain barrier, which suggests that carnitine can also be transported out of the brain [59]. The highest amounts of carnitine are especially high in the hypothalamus. The levels of carnitine acetyltransferase (CAT), an enzyme that synthesises acetyl-L-carnitine (ALC), are high in the hippocampus, colliculi and basal ganglia, and levels of carnitine palmitoyltransferase (CPT), which carries longer-chain fatty acids, are the highest in the hypothalamus. The ratio of ALC to FC is the highest in the brain (1:2) compared to cardiac muscle (1:4) and kidneys (1:10) [60]. Carnitine, in addition to its crucial contribution in long-chain fatty acid degradation, has an antioxidant effect, and as ALC, it has a neuroprotective and regenerative action on brain tissue [61,62]. Mitochondria, as the main producer of reactive oxygen species (ROS) during the energy generation process, and antioxidants also play key roles in the regulation of cell energy metabolism-dependent processes such as apoptosis, detoxification, regulation of membrane potential, Ca^2+^ buffering and distribution of ions in appropriate compartments of the cell, etc. Therefore, supporting proper mitochondria metabolism, especially in old age, is important for the correct functioning of nerve cells. Glucose is the main energy source for neurocytes. In a shortage of blood sugars in the brain, e.g., during fasting and starvation, long-chain fatty acids may replace sugars as energetic materials. However, in order to be utilised, a mitochondrial matrix must be reached with the help of carnitine [59]. LC plays an important role in the metabolism of brain lipids, transporting long-chain fatty acids into the mitochondrial matrix and it also transports acetyl groups from the mitochondrial matrix to the cytoplasm [16,59]. Therefore, LC and ALC function in fatty acid metabolism, ketosis and maintaining the proper concentration ratio of acyl-CoA to free CoA, which is important for proper metabolism in neurons [59]. Following the transfer of long-chain fatty acids into the mitochondria, LC reacts with free coenzyme A (CoASH) via carnitine palmitoyltransferase II (CPT II) to release LC [58].

By introducing long-chain fatty acids into the brain cells’ mitochondria, ALC allows reducing glycolysis and increases the use of alternative energy sources such as fatty acids and ketone bodies [59,63]. ALC may supply an acetyl moiety to brain cells mitochondria, or for the synthesis of acetylcholine (important neurotransmitter) [21,60], glutamate, glutamine and γ-aminobutyric acid (GABA), as well as brain lipids [61]. An indirect analgesic pathway has been shown to be associated with ALC because ALC reduces glutamate concentrations in the synapses. Glutamate—an excitatory neurotransmitter in the central nervous system—and its receptor—mGlu—are among the regulators involved in pain transmission [64]. ALC also increases the levels of the other neurotransmitters, i.e., noradrenaline and serotonin [65], and stimulates the activity of enzymes in the tricarboxylic acid cycle (TCA) and enzymes responsible for obtaining energy from fatty acids and amino acids. Moreover, ALC improves the metabolism of the brain cells’ mitochondria, the biosynthesis of proteins and phospholipids, has an anti-apoptotic effect, modulates the expression of gene-encoding brain proteins and protects neurocytes against toxic factors [64]. ALC stimulates the α-secretase activity involved in the cleavage of beta amyloid precursor protein (APP). In particular, ALC stimulates transport to the post-synaptic compartment of disintegrin and metalloproteinase-domain-containing protein 10 (ADAM10)—very important in alpha secretase activity and positively modulates its activity directed to APP—which are the most important proteins in the pathophysiology of Alzheimer’s disease [66]. AD patients showed a decrease in the growth-associated protein 43 (GAP-43) content in brain tissue and increased levels in CSF, which may be due to the degeneration of synapses in these patients. The function of GAP-43 is related to development, functional modulation of neural connections (synaptic plasticity) and regeneration of axons. Palmitoylcarnitine, favours the synthesis of GAP-43, as a component of the presynaptic membrane of neurons in the cortical and subcortical regions of the brain [67]. ALC protects neural stem cells by reducing apoptosis and restoring their proliferation. Namely, ALC decreases the expression of phosphatase and tensin homolog (PTEN), Bax, cytosolic cytochrome C and cleaved caspase-3 and -9, which are linked to death, and increases the expression of survival-related proteins, such as phosphorylated serine/threonine kinase Akt (pAkt) (Ser-473), phosphorylated glycogen synthase kinase 3b (pGSK3b), B cell lymphoma 2 (Bcl-2) and a nuclear nonhistone protein Ki-67 [68]. Another role of ALC is to stimulate the action of brain-derived neurotrophic factor (BDNF), which is reduced in AD patients [23,69]. ALC also elevates glutathione (potent antioxidant) concentration in astrocytes, which is reduced with age. Numerous studies have reported beneficial influence of ALC on the neuronal dopamine action because ALC slows down the progressive degradation of dopaminergic receptors and, at the same time, increases the level of dopamine in neurons (an important neurotransmitter responsible for thinking processes, coordination of movement, mood and resistance to stress). It was reported that LC supplementation of experimental animals significantly increases levels of neurotransmitters such as noradrenaline, adrenaline and serotonin, especially in brain regions rich in cholinergic neurons, i.e., the brain cortex, hippocampus and striatum [25].

Disturbances in L-carnitine biosynthesis and metabolism in AD patients have been documented. Cristofano et al. [70] demonstrated a reduction in the serum levels of free carnitine, acetyl-L-carnitine and other acylcarnitines in AD patients and concluded that reduced serum ALC level may predispose to AD and may contribute to neurodegeneration. Moreover, they found a progressive decline in ALC and other acyl-carnitine serum levels in subjects with subjective memory impairment (SMC) and mild cognitive impairment (MCI) down to the AD groups. The concentration of ALC significantly decreased by an average of 21% in SMC, 27% in MCI and 36% in AD compared to healthy subjects [70]. Therefore, ALC supplementation is beneficial to protect against Alzheimer’s and Parkinson’s diseases; to delay senile depression and age-related memory disorders; and ameliorate the work of the brain and nervous system, learning and memory, as confirmed in human clinical trials [62,71,72].

Providing carnitine with food, supplementation with carnitine or its derivatives is recommended to improve brain and nervous system action, the speed of learning and memorising, the level of brain energy, psychical condition as well as the effects of therapies in neurodegenerative disorders and peripheral neuropathies [18]. It was reported [72] that supplementation of adult mice with a mixture containing α-lipoic acid, acetyl-L-carnitine, docosahexaenoic acid. glycerophosphocholine and phosphatidylserine reduced reactive oxygen species by 57%. In addition, it was shown that supplementation with the above agents prevented a significant deterioration in cognitive functions and produced better neuroprotection. The broad cytoprotective properties of acetyl-L-carnitine in response to heat shock may be helpful in the treatment of tissue-damaging diseases such as neurodegeneration. It was confirmed that the treatment of rat astrocytes with acetyl-L-carnitine induces heme oxygenase-1 and that this effect was associated with the upregulation of heat shock protein 60, as well as a high expression of the redox-sensitive transcription factor Nrf2 in the nuclear fraction of treated cells [73]. ALC supplementation is used to treat neuropathy and neuropathic pain. The use of ALC in therapy is supported by biochemical and molecular studies, which prove its beneficial role in the modulation of neurotransmission and energy production [20]. The presented share of carnitine and acetyl-carnitine in the cellular metabolism of various biomolecules indicates its important role in the proper functioning of the nervous system (Table 2).

## 6. Lifestyle, Physical Exercise and Mental Activity in the Prevention of AD

There is growing evidence that three lifestyle components: social, mental and physical are inversely correlated with risk of dementia and Alzheimer’s disease. Cognitive disability associated with neurodegenerative diseases, in particular, AD, is becoming an increasingly serious cause of concern due to the dramatic increase in its incidence. Therefore, maintaining high physical, mental and social activity is one of the factors contributing to improving the quality and expectancy of life of elderly people, thereby reducing the risk of dementia and cognitive impairment in AD [32]. The use of various strategy programmes to prevent dementia, including the simplest, as for example, performing basic tasks related to memory stimulation, such as creating and realising a weekly plan or a schedule of activities during the day (practising simple physical exercises, reading newspapers, cooking, cleaning, meeting other people), counteract dementia and thus lead to a slowdown in cognitive decline [74]. It was reported that multi-domain programmes including diet, exercise, cognitive training and social activities give better results in slowing down disease progression than single-domain programmes [75]. Therefore, activating seniors to undertake education (Third Age Universities, computer courses), promoting volunteer initiatives (in hospitals, hospices, shelters), participation in social life, maintaining close relationships as well as promoting a healthy lifestyle delay dementia in neurodegenerative diseases and also improve cognitive function in people with AD [74]. In contrast, loneliness, as well as a lack of mental well-being and life satisfaction, are reflected as a worse quality of life in people with AD [76]. A great deal of evidence from numerous observations has shown that one third of AD cases worldwide are associated with seven common modifiable risk factors, i.e., diabetes, middle-aged hypertension, middle-age obesity, physical inactivity, depression, smoking and a low level of education [77]. Therefore, to reduce the likelihood of cognitive disability late in life, a high quality of life should be introduced at an early stage of life.

### 6.1. Physical Activity

Physical activity is an important factor in preventing pathological changes in dementia diseases, including AD. It is believed that physical exercises (morning gymnastics, walking, swimming, gardening, walking up the stairs and housework) intensify the formation of neurotrophic factors that are involved in maintaining homeostasis of the central nervous system and the regeneration of damaged brain tissue, thereby preventing destructive changes in the brain. In addition, physical exercise triggers the production of neurotrophins in the brain, which promote brain neuroplasticity and improve the functioning of neurons, also in those parts of the brain that are responsible for memory and other cognitive functions. Moreover, physical exercises stimulate the release of hormonal factors affecting neurons, such as brain-derived neurotrophic factor (BDNF) and epinephrine [78]. In addition to BDNF, insulin-like growth factor 1 (IGF-1) and vascular endothelial growth factor (VEGF) are also involved in the mechanisms by which exercise supports learning (regulated by IGF-1 and BDNF), but likewise, it supports normal neurogenesis and angiogenesis in the hippocampus—it appears to be regulated by IGF-1 and VEGF. These factors, produced in the periphery, penetrate the blood–brain barrier and lead to increased neuronal proliferation and differentiation [79]. Physical activity helps maintain optimal cardiovascular function, improves local blood flow and stops stroke and small blood vessel diseases. A comparative analysis of people exercising intensively with people who did not regularly do sports showed that those who exercised had a significantly lower risk of developing symptoms of AD disease within five years compared to people who did not do sports [80]. Another study by Paillard-Borg et al. [81] showed a 17-month difference in the occurrence of dementia between an inactive group and the most active group. The highest risk of cognitive impairment was observed in people without physical, mental or social activity. Therefore, it is believed that physical activity protects against cognitive impairment in the elderly and slows down the course of AD disease, thus delaying the onset of dementia [82]. In another cohort study, reported by Law et al. [83], it has been shown that in middle-aged adults at risk of cognitive disease, moderate physical activity (but not light or intensive) has a beneficial effect on CSF biomarkers (higher Aβ42, lower total tau/Aβ42 and lower phosphorylated tau/Aβ42). In contrast, a sedentary lifestyle was associated with reduced Aβ42 in the CSF of the study group. It is, therefore, believed that physical exercise and cognitive training prevent cognitive memory deficits associated with β-amyloid neurotoxicity [84,85]. Aerobic exercise, including morning gymnastics, cycling, swimming, walking, running, rollerblading, skipping or cross-country skiing, as well as cognitive training (aerobic fitness) improve blood supply and thus brain oxygenation. Thanks to the combination of aerobic exercises and aerobic fitness a synergistic effect on increasing cognitive functions, by improving the structure of the brain and its functioning, may be achieved [86].

Although the general pathogenesis of AD is being clarified, the exact mechanisms of AD’s pathogenesis are still unclear. In recent years, it has been reported that impaired autophagy (or autophagocytosis) associated with microRNA (miRNA) dysfunction is involved in ageing and neurodegenerative diseases. Therefore, the regulation of autophagocytosis via miRNA can become one of the strong AD intervention strategies. In addition, autophagy is tightly regulated by the signalling pathway of mTOR (mechanistic target of rapamycin, mammalian target of rapamycin), which is a serine-threonine kinase that regulates the rate of some intracellular processes in response to extracellular signals. mTOR in the central nervous system plays multiple roles: regulates cell viability, differentiation, transcription, translation, protein degradation, ribosome biogenesis, actin cytoskeletal organisation and autophagy, as well as the development of axonal and dendritic trees, synaptogenesis, synaptic plasticity and learning and memory. Dysregulation of the mTOR kinases pathway can be one of the causes of numerous neuropathologies and neurodegenerative diseases, including Alzheimer’s, Parkinson’s and Huntington’s diseases. Recent studies have shown that physical activity helps to regulate the state of autophagy via the mTOR signalling pathway, which can prevent and be helpful in treating AD [87]. In addition, physical activity and physical fitness improve cerebral blood flow (CBF) in the lower temporal gyrus, angular veins and posterior gyrus (reduced CBF is a hallmark of ageing and AD disease), which has been proven to be the predictor preceding cognitive impairment in the elderly [26].

### 6.2. Mental Activity

Dementia is a syndrome characterised by cognitive disorders such as the impairment of memory, abstract thinking, orientation, understanding, counting, learning ability, language functions and the ability to compare, evaluate and make choices. As a result of the above-presented developing disorders, intellectual performance is reduced, and efficient functioning in everyday life is impaired. Alzheimer’s disease and/or damage to the cerebral vessels with impaired blood flow are the most common causes of dementia, where many patients acquire it simultaneously. There is scientific evidence that the factors associated with a healthy lifestyle, such as regular exercise, high mental activity, a healthy diet, non-smoking, maintaining normal blood pressure and cholesterol, avoiding depression and middle-aged overweight/obesity and proper control of diabetes (if present), help reduce the risk of dementia [88,89].

Preventive measures against the onset of neurodegenerative diseases (including AD) should start early, before the occurrence of pronounced structural changes in the brain, and not in old age. In a Finnish multimodal intervention study consisting of diet, exercise, cognitive training and social activity, vascular risk monitoring concerned the prevention of cognitive impairment and disability (FINGER—Finnish Geriatric Intervention Study to Prevent Cognitive Impairment and Disability—it is a multicentre, randomized, rigorously controlled trial) in people aged 60–77 years with an increased risk of dementia (but without dementia/significant cognitive impairment) after 2 years of experiments between the study and control group (general health advice). The study group was included in a balanced diet and exercise intervention. The cognitive training included group activities and individual sessions. Individual sessions consisted of computer-based training. The cognitive training included executive processes, i.e., updating spatial, updating letter, updating number and mental set shifting tasks; working memory, i.e., maintenance tasks; episodic memory, i.e., relational and spatial tasks; and mental speed, i.e., shape match task. Social activities were stimulated through the group meetings. Observational outcomes of cognitive function were measured by the modified Neuropsychological Test Battery, Stroop test and Trail Making Test [90]. This study noted significant intervention effects on the overall cognition and executive functioning and processing speed and other outcomes, i.e., BMI, dietary habits and physical activity. No significant effect was noted on memory, but post-hoc analyses showed an effect on more complex memory tasks and also beneficial effects on lower risk of cognitive decline [90]. This study supports the efficacy of multidomain prevention approaches. Other authors in the FINGER study [91] investigated the relationship between change in CAIDE (Cardiovascular Risk Factors, Aging and Dementia) score and change in neuroimaging biomarkers—MRI (Magnetic Resonance Imaging) and Pittsburgh Compound B-positron emission tomography (PiB-PET). The participants had brain measurements (hippocampal, total grey matter and white matter lesion volumes and Alzheimer’s disease signature cortical thickness). A reduction in the CAIDE score was observed in 30% participants in the intervention and 21% in the control group. In this neuroimaging study, a reduction in the CAIDE score (indicating lower dementia risk) during the intervention was associated with lesser decline in hippocampal volume. It is suggested that preventive strategies may be more effective if started early, before the more pronounced structural changes in the brain that characterise Alzeimer’s disease develop. According to the authors, it is important to develop dementia risk assessment tools that will be even more sensitive to lifestyle changes and their potential effects on brain structure [84,90,91]. The large-scale online programmes on healthy lifestyles and brain health enable the implementation of preventive measures in a healthy adult population. Implementing these programmes can help prevent dementia in old age [92]. Primary prevention of dementia aims to reduce the risk factors by focusing efforts on improving the lifestyle of middle-aged people before or in the earliest stages of neuropathological changes that characterise AD and other types of dementia. An alternative strategy is secondary prevention, which aims to minimise the symptoms of progressive disease, characterised by subjective cognitive decline and mild cognitive impairment [93]. All forms of memory training activities by seniors, such as solving crosswords, reading books and magazines; arranging puzzles; playing cards, checkers, chess and board games; and participation in musical classes, are a way to maintain intellectual activity and mental fitness. Participation in special programmes of cognitive activity, organised by various Memory Disorders Therapy Centres, which help improve attention, concentration, perceptiveness, logical thinking, learning ability and visual memory, i.e., cognitive functions, is necessary for the proper functioning of the elderly [94,95].

## 7. Conclusions

The presented review of literature emphasises the importance of the early prevention of cognitive decline in Alzheimer’s disease by promoting physical, intellectual, social activity and a proper diet. Daily physical activity and a proper diet are the key to a long, healthy life. The low to moderate intensity exercise for both younger and older people is necessary to maintain good physical and mental form and prevent the risk of many diseases, including neurodegenerative diseases. In the current food pyramid for healthy lifestyle, physical activity has the most important place. We should limit our sedentary lifestyle and engage in physical activity as often as possible, not only as a typical sport, but also walking on stairs, just walking, Nordic walking, swimming, stepper and team games. Based on many studies, it can be stated that in preventive recommendations, physical exercises have a protective effect on cognitive functions and general fitness of people of all ages and in patients with AD. The latest reports indicate that the most effective method of non-pharmacological intervention in cognitive disorders in old age is multimodal therapy, i.e., a combination of intellectual stimulation, social activity and physical exercise.

According to dietary recommendations, increased consumption of fish fats, vegetable oils, low-starchy vegetables, fruits with a low glycaemic index and a diet with low sugar content as well as moderate wine/beer consumption reduce the risk of dementia, including AD. In contrast, red meat, eggs, dairy and marine fish rich in choline, lecithin and carnitine are potential sources of the harmful toxic TMAO. Thus, the reduced dietary intake of TMAO precursors can lower its levels. L-carnitine, in addition to its crucial contribution in long-chain fatty acids degradation, has many other actions, including an antioxidant effect, and as acetyl-L-carnitine, it has a neuroprotective and regenerative action on brain tissue; therefore, reasonable consumption of carnitine-rich foods or L-carnitine supplementation seems appropriate.

To reduce the risk of cognitive decline, based on the presented evidence, we suggest the following measures: physical and mental activity; avoidance of infection and stress; the Mediterranean diet; a high level of consumption of mono- and polyunsaturated fatty acids and limiting the consumption of saturated fatty acids; increasing the consumption of fruit and vegetables; a lower intake of full-fat dairy products and/or dairy fats; and calorie restriction.

Undoubtedly, a key aspect of a healthy lifestyle is a common-sense approach to nutrition that allows a person to make informed dietary choices along with activity can prevent the onset of AD or reduce the severity of AD-related symptoms. Most people are still unaware of the relationship between lifestyle and brain health, which indicates a need for public health campaigns. Raising awareness in the general population about lifestyle modification is a key factor in preventing dementia in neurodegenerative diseases, including AD.

To sum up, physical and intellectual exercise and diet (Mediterranean, low-calorie, rich in antioxidants) strengthen defence mechanisms and slow down the ageing process of the body.

## Figures and Tables

**Table 1 nutrients-14-01534-t001:** Nutritional pyramid and physical activity recommendations for healthy adults [27,28,29,30,31,32], modified.

Order Pyramid	Food Group	Recommended	Avoid	Comments
1.	Physical exercise	Minimum 30–45 min/day.	Limit your sedentary lifestyle.	Cycling, swimming, walking, gardening, walking up the stairs and housework.
2.	Fluids	Water (non-carbonated, mineral medium or highly mineralised), tea, coffee, fresh fruit and vegetable juice.	Limit drinking boiled or sparkling water, limit the consumption of sweetened drinks and flavoured waters.	Drink water (about 1.5–2 L/day). Provide water regularly, in small portions throughout the day. A glass of water should be drunk immediately after waking up. Drink water between meals (1 glass at least 15 min before meals and 15 min after meals). Pure mineral water should be the main source of hydration.
3.	Vegetables and fruit	Preferably raw or briefly cooked.	Limit the sugar and sweets from the diet (replace them with fruit and nuts, pumpkin seeds, sunflower).	Minimum 400 g of vegetables and fruit divided into 5 portions (one portion = one cup). They should constitute a minimum of 50% of the daily portion of food (one portion may be a glass of freshly squeezed juice). Remember the right proportions: vegetables should make up the majority, about three-quarters, and fruits —one-quarter. Important variety.
4.	Grain products	Whole-grain food products (whole grain brown rice, whole-wheat noodles and cereal groats including buckwheat and barley).	Do not consume highly processed products.	At least half of all cereals consumed should be whole grains. Dietary fibre regulates the functioning of the digestive tract, facilitates the maintenance of normal body weight, prevents constipation and the formation of colon cancer, reduces the content of cholesterol in the blood.
5.	Milk and milk products	Milk (with up to 2% fat), yogurt, kefir, buttermilk and partly cottage cheese.	Avoid ready-made, flavoured dairy products with additional flavour ingredients (sugar, aromas and dyes).	Minimum 2 glasses/day (or other dairy drinks) and partially with cheese, e.g., 1 cup (200 mL) of kefir / yogurt, or 280–400 g semi-skimmed cheese, or 1 slice (30 g) of yellow cheese. The rennet cheeses should be consumed less often (due to their higher fat and higher energy content).
6.	Meat and meat products	Fish (salmon, tuna, herring, mackerel, cod), poultry, lean meat (ham, sirloin, fillet, pork loin).	Limit meat consumption (especially red and processed meat products to 0.5 kg/week). Avoid eating meat preparations —they contain a large amount of salt, phosphates, nitrites, water, dyes, aromas, flavour enhancers, sugar, starch, soy protein and other additives with a relatively low meat content.	Meat substitutes, rich in protein, are eggs, legumes (beans, lentils, peas, soybeans); it is worth eating them 1-2 times a week. The meat should be processed as little as possible, preferably cooked, stewed without frying or baked in foil or ovenproof dish.
7.	Vegetable oils	Oils: olive, canola, soybean, sunflower, peanut and other vegetable oils and margarines without trans fatty acids.	Animal fats.	Replace animal fats with vegetable oils, nuts and seeds. Consume in small amounts and preferably in raw form, as an addition to salads or other dishes. For short-term frying, use rapeseed oil or olive oil. Deep frying: saturated and monounsaturated fats (lard, clarified butter, coconut oil).
8.	Herbs	Fresh and dried.	Prepared spice mixtures.	Use herbs/spices such as rosemary, oregano, thyme, basil, turmeric, garlic, ginger and cinnamon on a daily basis. Herbs and other natural spices improve taste and have valuable ingredients, including antioxidant properties.
9.	Salt (NaCl)	The salt substitutes—potassium or magnesium salt. Natural spices and herbs instead of salt	Limit the addition of salt to food, to consumption during cooking and preparation.	Salt (including products, e.g., bread, sausages, cheese, salty snacks and salting-out) should be consumed in an amount of not more than 5 g/day (approximately a flat teaspoon). Use rock and iodised salt. Limit the consumption of foods containing large amounts of sodium: meats, canned meat and fish; rennet and blue cheese; silage; smoked products; marinated vegetables; soups and powdered sauces; spice mixtures; broth cubes; salty snacks (chips, sticks, pretzels, crackers, peanuts).
10.	Sugar	Can be replaced by fruit and nuts, brown sugar (unrefined), natural sweeteners, i.e., stevia, xylitol, maple and date syrup, honey.	Limit the consumption of white sugar, synthetic sweeteners, sweets.	Limit to 10% of total energy: less than 10% of 2000 kcal = 200 kcal = equivalent to 10 teaspoons of sugar (50 g). Keep your intake of sugar, sweeteners, added sugars and naturally occurring sugars in fruit juices and honey in moderation.
11.	Alcohol	Beer/non-alcoholic beer, wine.	Allowed in moderate amounts. Reducing heavy alcohol use may be an effective dementia prevention strategy.	There are no precisely formulated clinical recommendations for alcohol consumtion. People choosing alcoholic beverages must do it with caution and moderation.

**Table 2 nutrients-14-01534-t002:** Main pleiotropic effects of L-carnitine (LC), acetyl-L-carnitine (ALC) and palmitoylcarnitine on the central nervous system.

Course of Action	Effect of Action	Author, Year, Ref.
Neuroprotective, neurotrophic, neuromodulatory effect	ALC accelerates regeneration of neurons; improves the synthesis, stabilisation, fluidity and functionality of neuronal membranes; accelerates protein synthesis; and improves the axonal transport of neurofilament proteins and correct tubulin acetylation. ALC improves brain-derived neurotrophic factor (BDNF), elevates levels of nerve growth factor (NGF) (neurotrophic factors that regulate, among other things, survival, differentiation, maturation, synapse formation and neuronal growth). ALC favours the delivery of disintegrin and metalloproteinase 10 (ADAM10), which is the main α-secretase that cleaves amyloid precursor protein (APP). ALC downregulates the expression of the myelin basic protein (MBP) and the ATP synthase lipid-binding protein, subunit c genes. ALC upregulates the expression of brain-specific Na^+^-dependent inorganic phosphate transporter gene and prostaglandin D2 synthase gene (PGD2S), protecting against excitotoxic injury. ALC reduces apoptosis and restores proliferation by increasing expression of survival-related proteins, such as phosphorylated Akt (pAkt), phosphorylated glycogen synthase kinase 3b (pGSK3b), B cell lymphoma 2 (Bcl-2), Ki-67 protein. ALC reduces the expression of death-related proteins, such as Bax, cytosolic cytochrome C, cleaved caspase-3 and -9. Palmitoylcarnitine, favours synthesis of growth-associated protein 43 (GAP-43), a component of the presynaptic membrane of neurons in the cortical and subcortical regions of the brain, which is related to development, functional modulation of neural connections (synaptic plasticity) and regeneration of axons.	Ribas et al., 2014, [62]; Respondek et al., 2015, [69]; Zhou et al., 2011, [23]; Virmani & Binienda, 2004, [60]; Ferreira & McKenna, 2017, [61]; Traina, 2016, [19]; Epis et al., 2008, [66]; Sergi et al., 2018, [64]; Bak et al., 2016, [68]; Nałęcz et al., 2004, [67]; Bak et al., 2016, [68]
Participation in the regulation of the energy metabolism of the brain	ALC allows reduces glycolysis and increases the use of alternative energy sources such as fatty acids and ketone bodies. ALC stimulates the activity of enzymes in the TCA cycle when metabolism via the pyruvate dehydrogenase (PDH) complex is impaired (as occurs in hypoxia and traumatic brain injury). ALC increases the activity of α-ketoglutarate dehydrogenase (α-KGDH) in intrasynaptic but not in non-synaptic mitochondria from cerebral cortex. ALC increases the levels of pro-glycogen (glycogen precursor). ALC stimulates glucose metabolism and utilisation by the use of the acetyl-CoA from ALC. ALC lowers the level of lactate that builds up in the brain during ischemia.	Jones et al., 2010, [59]; Aureli et al., 1998, [63]; Ferreira & McKenna, 2017, [61]; Traina, 2016, [19]
Antioxidative, anti-inflammatory effects and other metabolic functions	LC has the ability to scavenge to hydrogen peroxide and superoxide radicals as well as chelate transition metal ions. LC and ALC increase reduced glutathione (GSH) levels in astrocytes and GSH/GSSG ratio. ALC decreases levels of reactive nitrogen species and protein nitration. ALC protects cells against oxidative injury, induces of heme oxygenase-1 (HO-1), which has antioxidant, anti-inflammatory, anti-apoptotic, cytoprotective effect, and ALC upregulates the expression of heme oxygenase-1 gene (HMOX1), reducing the amyloid-beta toxicity. ALC inhibits activity of inducible nitric oxide synthase (iNOS). ALC increases cytochrome b content and cytochrome bc1 complex, increasing the activity of electron transport chain (ETC), complexes and stimulates the oxidative phosphorylation. ALC upregulates cytochrome c oxidase (COX). ALC upregulates the expression of the lysosomal H+/ATPase, V1 subunit of D gene. LC and ALC induce the elevation of the heat shock protein (HSP) levels and the activation of phosphoinositol-3 kinase (PI3K), protein kinase G (PKG) and proline-directed kinases (ERK1/2) pathways, which play essential roles in neuronal cell survival, thereby inducing the expression of anti-apoptotic and anti-oxidant proteins.	Juliet et al., 2003, [25]; Ribas et al., 2014, [62]; Traina, 2016, [19]; Ferreira & McKenna, 2017, [61]; Calabrese et al., 2005, [73]
Participation in the metabolic processes of fatty acids	LC and ALC allow the transport of acetyl groups between different intracellular compartments. ALC modulates the activity of neurons by increasing the synthesis of phospholipids necessary for membrane formation and integrity. It plays a role in membrane repair by reacylation of phospholipids. Turnover of lipids containing carbons from ALC and reutilisation of the carbons for myelination and cell growth. LC and ALC are involved in the metabolism of fatty acids, ketosis, which are the basic energy substrates for the brain in conditions of metabolic disturbances (fasting or starvation).	Jones et al., 2010, [59]; Virmani & Binienda, 2004, [60]; Ferreira & McKenna, 2017, [61]; Traina, 2016, [19]
Influence on the level and activity of proteins (receptors) and neurotransmitters	ALC increases the secretion of dopamine in neurons. ALC is a potential source of acetyl groups for the synthesis of acetylcholine. The acetyl moiety of ALC is incorporated into the carbon skeleton of the gamma-aminobutyric acid (GABA) and glutamate. ALC increases the levels of serotonin and noradrenaline in the hippocampus and cortex. ALC enhances the transcription of the GRM2 gene, encoding the metabotropic glutamate receptor type-2 (mGLU2), through the acetylation and activation of nuclear factor kappa B (NF-kB p65). In nerve injury, the mGlu2 receptor overexpressed by ALC binds the glutamate, reducing its concentration in the synapses with an analgesic effect. ALC stimulates α-secretase activity and physiological amyloid precursor protein (APP) metabolism. ALC upregulates kinesin light chain 1 (KLC1) gene expression in the brain of AD patients, thereby reducing the deposition of amyloid precursor protein (APP) in the brain. ALC has the ability to acetylate the lysine 28 of the Aβ peptide of β-amyloid, thereby reducing the formation of toxic β-sheet aggregates of β-amyloid. ALC increases the expression of neurotrophin receptor p75-mRNA level, which plays important roles in regulating β-amyloid metabolism in the brain. ALC modulates, directly or indirectly, the N-methyl-D-aspartate (NMDA) receptor, which introduces Ca^2+^ into neurons.	Wite & Scates, 1990, [21]; Virmani & Binienda, 2004, [60]; Ferreira & McKenna, 2017, [61]; Sergi et al., 2018, [64]; Smeland et al., 2012, [65]; Chauhan et al., 2003, [22]; Calvani et al., 1992, [24]

## Data Availability

Not applicable.

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
