# Peer review of "Healthy Food Pyramid as Well as Physical and Mental Activity in the Prevention of Alzheimer’s Disease"

_nutrients, 2022, doi:10.3390/nu14081534_

Round 1

Reviewer 1 Report

To increase awareness of preventive measures of AD due lifestyle is of outmost importance, thus your article is important. However,  I have some conserns about the article,  Please find below my commets:

Major:

  • The article lacks some precision and some very important studies in the field are missing or not correctly referenced (secondary references used, main outcome not understood). It is of outmost importance to correct this before possible publication
  • Even though Mediterranean diet is described in the article, the landmark RCT study is not referenced Martínez-Lapiscina EH, Clavero P, Toledo E, Estruch R, Salas-Salvadó J, San Julián B, Sanchez-Tainta A, Ros E, Valls-Pedret C, Martinez-Gonzalez MÁ. Mediterranean diet improves cognition: the PREDIMED-NAVARRA randomised trial. J Neurol Neurosurg Psychiatry. 2013 Dec;84(12):1318-25. doi: 10.1136/jnnp-2012-304792.
  • In my opinion there should be a subtitle, where studies combining diet and exercise should be added. Now for example Finger study is under headline mental activity, even though the main emphasis on the study was diet and exercise
  • Another important study combining aerobic exercise and DASH diet is missing entirely: Study of Blumenthal et al. Blumenthal JA, Smith PJ, Mabe S, et al. Lifestyle and neurocognition in older adults with cognitive impairments: A randomized trial. Neurology. 2019;92(3):e212-e223. doi:10.1212/WNL.0000000000006784.
  • Lines 653-660; the Finger trial is not correctly described or referenced in the article. This is one of the key studies in the field and it should absolutely be correct in the article! “The authors don’t use the original RCT trial as a reference, and indicate that “Finnish multimodal intervention study (diet, exercise,cognitive training, vascular risk monitoring) about the prevention of cognitive impairment and disability (FINGER) in people aged 60-77 years with an increased risk of dementia (but without dementia/significant cognitive impairment), did not show significant positive changes in MRI (regional brain volumes, cortical thickness and white matter lesion volume) after 2 years of experiments between the study and control group. However, post hoc analysis showed a beneficial effect of multimodal intervention on cognitive functions in the study group [84,90].”
  • The improvement of cognitive domains were the main outcomes of the RCT, not post-hoc analysis
  • brain volumes on the other hand were not the main outcome of the trial

The original reference: Ngandu T, Lehtisalo J, Solomon A, Levälahti E, Ahtiluoto S, Antikainen R, Bäckman L, Hänninen T, Jula A, Laatikainen T, Lindström J, Mangialasche F, Paajanen T, Pajala S, Peltonen M, Rauramaa R, Stigsdotter-Neely A, Strandberg T, Tuomilehto J, Soininen H, Kivipelto M. A 2 year multidomain intervention of diet, exercise, cognitive training, and vascular risk monitoring versus control to prevent cognitive decline in at-risk elderly people (FINGER): a randomised controlled trial. Lancet. 2015 Jun 6;385(9984):2255-63. doi: 10.1016/S0140-6736(15)60461-5. 

  • There are many more recent studies on cognition, diet and exercise

I find it very confusing in this article that there is so much emphasis on L-carnitine and its’ derivates, although there is limited evidence of the supplement’s effect to prevent AD. There are many other compounds and nutriceuticals also, but they are completely left out of this article. The title of the article is “Healthy food pyramid as well as physical and mental activity in the prevention of Alzheimer's disease”, it does not go with the purpose of the article to put so much emphasis on one supplement/intake from foods. The emphasis of the article should be according to the title. Huge table on L-carnitine should be under another article title completely. If one is writing an article on nutriceuticals or combounds that may benefit brain health, that would be ok, but it does not sit well with the theme.

Vitamin B12 is only mentioned briefly in the article. It is well known that some older people may have problems with absorption of the vitamin and that may increase their risk of AD and decline cognition.

I was wondering about cocoa, tea and coffee. They may also have some beneficial effects on cognition, but they were left out completely.

Conclusion:

Statement “moderate wine/beer consumption, reduce the risk of dementia” is false. This can not be backed up by science. It is true that low amounts of alcohol are part of Mediterranean diet, but there is not plausible scientific evidence that could make such a claim. Generally, moderate alcohol consumption is not good for brain.

Minor

Other causes that increase risk of AD, lines 95-97

hearing problems

aging

high alcohol consumption

metabolic disease

high cholesterol levels

deficiency of B-vitamins

You mention nutriceuticals, but fail to present a recent landmark study on the subject Lipiddiet- study, that got good results; Soininen H, Solomon A, Visser PJ, Hendrix SB, Blennow K, Kivipelto M, Hartmann T; LipiDiDiet clinical study group. 36-month LipiDiDiet multinutrient clinical trial in prodromal Alzheimer's disease. Alzheimers Dement. 2021 Jan;17(1):29-40. doi: 10.1002/alz.12172. Epub 2020 Sep 13. Erratum in: Alzheimers Dement. 2021 May;17(5):909.

  1. 3 line 109

Use of inclusive language is encouraged; instead of elderly; use older people, older adults etc.

  1. 3 line 112

polyphenols may also have other health promoting roles than merely functioning as antioxidants; such as vasodilation of cocoa just for an example

  1. 3, lines 117-123

“Therefore, the nutritional recommendations for older people include primarily eating vegetables, fruits, cereals, legumes, vegetable oils, and fishes, whereas dairy products and meat should be consumed in moderate amounts because red meat, egg yolks, offal (brain, liver, heart, kidneys), wheat germ, yeast, legume seeds, nuts, dairy and marine fish rich in choline, lecithin and L-carnitine are potential sources of the harmful toxic trimethylamine N- 122 oxide (TMAO) that can be converted into the carcinogenic N,N-Dimethyl-N-Nitrosoam-ine (NDMA) [15]

This is a very weird sentence; many nutritional recommendations for older people encourage people to eat dairy for its’ high protein content, because older people are in risk of muscle loss and low protein intake, nuts are generally recommended in all health promoted diets and they were a central component in the PREDIMED- diet, fish is generally considered an important part of healthy diet and it is an important part of a heart and brain healthy diet. Moreover, choline is also considered important for brain health.

Clarify, this sentence:  “Mediterranean and  Asian diets are very similar to each other and are considered the healthiest.” There is a lot of data on Mediterranean diet, but I have not seen a lot of studies on Asian diet being one of the healthiest. Rice eaten in Asian diet is usually white, processed rice. Use references to prove your claim.

Reviewer 2 Report

The manuscript entitled " Healthy food pyramid as well as physical and mental activity in the prevention of Alzheimer's disease" reviews the role of food and physical activity on mental health.

The manuscript is interesting will be a good resource for the reader on the effects of food and physical activity on Alzheimer's disease. 

The manuscript discuss and included so many facts about food and other activities without any citation. Citations are important and required to justify the facts. For example, the information included in Table 1 does not have any references.

Also, reactive oxygen species and antioxidant-containing food should be discussed in detail. ROS play important role in Alzheimer's disease.
